# Microsatellite Uncertainty Control Using Deterministic Artificial Intelligence

**DOI:** 10.3390/s22228723

**Published:** 2022-11-11

**Authors:** Evan Wilt, Timothy Sands

**Affiliations:** 1Sibley School of Mechanical and Aerospace Engineering, Cornell University, Ithaca, NY 14850, USA; 2Department of Mechanical and Aerospace Engineering, Naval Postgraduate School, Monterey, CA 93943, USA

**Keywords:** control systems, feedforward, feedback, adaptive control, deterministic artificial intelligence

## Abstract

This manuscript explores the applications of deterministic artificial intelligence (DAI) in a space environment in response to unknown sensor noise and sudden changes in craft physical parameters. The current state of the art literature has proposed the method, but only ideal environments, and accordingly this article addresses the literature gaps by critically evaluating efficacy in the face of unaddressed parametric uncertainties. We compare an idealized combined non-linear feedforward (FFD) and linearized feedback (FB) control scheme with an altered feedforward, feedback, and deterministic artificial intelligence scheme in the presence of simulated craft damage and environmental disturbances. Mean trajectory tracking error was improved over 91%, while the standard deviation was improved over 97% whilst improving (reducing) control effort by 13%.

## 1. Introduction

Attitude determination and control is a critical subsystem on the majority of modern spacecraft like that depicted in Figure 1. Rotational dynamics and kinematics, actuators, sensors and observers, controllers, and perturbations are all expansive areas of research into ensuring the desired pointing and spin of a given craft is achieved. Traditional linearized feedback control is anything but simple and robust. Extensive analysis of poles, gain design, and analysis on the boundaries of where linearization still holds true all pose consistent challenges even in simple implementations of traditional proportional-integral-derivative (PID). Large amounts of thought are put into designing controllers for their specific implementations and their performances are highly restricted to the anticipated environment for which they were designed. To address the limitations of linearization, non-linear feedforward techniques were developed. Though more robust than traditional feedback (and in theory capable of exact control), feedforward control also has innate restrictions. It requires highly accurate process modeling that is easily disturbed by unaccounted for noise. deterministic artificial intelligence conveniently ignores environmental noise and requires only knowledge of rotational dynamics. By assuming the physics will hold true (which it always will), we can create highly accurate and robust controllers. In conjunction with feedback and feedforward control, our new hybrid control can perform accurately in a wide range of environments and shifting variables.

Attitude control of space vehicles is complicated by external forces and torques and unfortunately sometimes collision damage due to coincident orbital events. Significant research in the topic follows several lines seeking to negate such complicating factors. A long list of techniques in the lineage include classical feedback, feedforward (more relatively rare), optimal control, robust control (typically a label for optimal control minimizing the infinity norm of specified cost function), and several nonlinear adaptive techniques like self-tuning regulators and model-reference adaptive control. Another recent technique is physics-based control. Several of these techniques necessitate autonomous formulation of attitude trajectories and several options predominate. Especially noting the utilization of the eigenaxis [3] by the four-dimensional parameterization of three-dimensional rotation (the “quaternion”), so-called eigen-axis maneuvers [4] utilize the eigen-axis to define the shortest-distance between two point (the initial point and the desired point). Subsequent demonstration that shortest distance does not lead to minimum time maneuvers, [5] alternative options for autonomous trajectory generation abound. Having the control follow a “sliding-mode” manifold [6,7] defined as by linearity with a nonlinear control forcing the trajectory to follow the sliding mode, where Zou et al., [8] also sought to introduce learning using a stochastic (non-deterministic) neural network. Wang et al., [9] sought to include actuator dynamics while Wang et al., [10] incorporated vehicle structural flexibility.

Nonlinear adaptive techniques established the provenance of self-awareness statements in deterministic artificial intelligence (later combined with physics-based controls). Slotine proposed nonlinear adaptive methods for robotics [11] with parallel development for spacecraft attitude control [12] culminating in the finalized development of both presented as advanced material in the textbook *Applied Nonlinear Control*. [13] The method essentially utilized classic feedback to adapt a feedforward and feedback signals in addition to adaptive the trajectory fed to both feedforward and feedback channels. The foundational work parameterized the control in the non-rotating, non-accelerating inertial reference frame, while Fossen [14] proposed improved performance (computational efficiency) with analytically identical control formulated in the rotating body reference frame. Fossen illustrated broad, general applicability to underwater robotics and underwater vehicles in general [15,16], and this generalization continued eventually culminating in similar demonstrations for the currently instantiation of deterministic artificial intelligence.

After Fossen’s evolutions, the feedforward and feedback gains were proven to be separately tunable [17,18], while the former works combined tuning of both. The evolving techniques were augmented by physics-based control methods of Lorenz [19] formerly applied to torque control [20] but also applied specifically to deadbeat torque generation, induction machines [21], multi-phase electric motors [22], magnetic state control [23], loss-minimizing servo control [24], magnetic flux control [25], self-sensing control [26], and brushed DC motor control [27] eventually illustrating efficacy for spacecraft attitude control [28] as an augmentation to the lineage of methods started by Slotine.

Regarding the feedforward channel, the most recent developments proposed methods for dealing with highly nonlinear oscillatory van der Pol circuits [29] necessitating state observers [30] to permit applicability in feedback [31] as learning by Smeresky and Rizzo to spacecraft attitude control. Following this success, very similarly to the earlier lineage, the methods were applied to unmanned underwater vehicles [32] and DC motor control [33] where an interesting comparison to foundational nonlinear adaptive control was offered by Shah [34]. So far, this year the current instantiation of deterministic artificial intelligence has been applied to remote underwater vehicles by Osler [35], while the necessary autonomous trajectory generation schemes currently in use were critically compared by Sandberg [36] following Koo’s elaboration [37] of the impacts on trajectories of discretization and numerical propagation. Sandberg’s trajectory generation results were enhanced by Raigoza [38] to include satellite de-orbiting with autonomous obstacle avoidance.

This considerable background literature display gaps emphasized strongly by the instantiation of the work presented here, and based on the major gaps of overcoming uncertainties, the claimed contributions of the article are justified. One major feature offered by the self-awareness statements of deterministic artificial intelligence obfuscates the necessity in the stochastic approaches to categorize uncertainties as internal or external, parametric or non-parametric, constant, characteristic or random. In real time applications the physical, mechanical, electrical and environmental constraints are addressable in real time environments by a priori utilization of real-time optimization techniques presented by Sandberg in [36].

Artificial intelligence is ubiquitously described as intrinsically stochastic, implying machine learning, most often utilizing neural networks and often augmented with deep learning. While this manuscript utilizes a number of uncertainties, disturbances and noises the algorithm remains deterministic in the assertion of self-awareness (offered by Cooper and Heidlauf’s methodology), relegating the stochastic forms of artificial intelligence to counteract the unknowable features.

Smeresky and Rizzo [31] showed that deterministic artificial intelligence could achieve improved results compared to both optimized feedback and feedforward control schemes. Additionally, it achieved reduced computational burden. Building off their work, this manuscript presents a redesigned deterministic artificial intelligence controller in conjunction with Smeresky’s optimal feedback and feedforward control and evaluates the proposed method to overcome a range of environmental disturbances (not formerly in the literature) including a sudden shift in the values of the craft’s mass moments of inertia, while the controller was tuned with no such knowledge. Despite this sudden shift and range of disturbances, the controller exceeded expectations and demonstrated consistent stability. Mean trajectory tracking error was improved over 91%, while the standard deviation was improved over 97% whilst improving (reducing) control effort by 13%.

## 2. Materials and Methods

Comparisons of the various algorithms should be performed under equal conditions. Therefore, the validating simulations displayed in Appendix A illustrate identical conditions where algorithms are switched on when active, while others are deactivated. The parameters of the respective algorithms are chosen seeking identical final values when initiated from identical initial conditions.

### 2.1. Rigid Body Motion

The topic of rigid body motion is best described in a Euler’s Equation for rigid body motion displayed in Equation (1) (reference Table 1 for variable definitions in Equation (1)).
(1)$$T=\dot{H}+\omega \times H_{S},$$

This is the rotational equivalent of newton’s second law relating the acceleration of an object to the fore applied to it. It is the basis of action from the controllers and actuators on the craft, determining the exact quantity of torque necessary to shift the angular momentum of the craft to a specific direction and magnitude.

### 2.2. PDI Control

Traditional PID control functions around a linearized control point as illustrated in Equation (2) (reference Table 2 for variable definitions in Equations (2) and (3)).
(2)$$u_{fb}=k_{p}e_{\theta }+k_{d}{\dot{e}}_{\theta }+k_{i}\int e_{\theta }.$$

We implement a non-linear enhanced control called proportional-integral-derivative (PDI) of the form displayed in Equation (3).
(3)$$u_{fb}=-k_{p}\left(\theta_{d}-\theta \right)-k_{d}\left(\omega_{d}-\omega \right)-k_{i}\int \left(\theta_{d}-\theta \right)dt-\omega \times J\omega ,$$

This controller accounts for both position and velocity errors, better reflecting the true physics of the non-linear system we are modeling. Additionally, it contains a non-linear decoupling term (ω×Jω) to account for the constantly shifting reference frames.

### 2.3. Luenberger Observers

We implement an observer of the form displayed in Equation (4) (reference Table 3 for variable definitions in Equation (4)).
(4)$$x\left(k+1\right)=A_{d}\hat{x}\left(k\right)+B_{d}u\left(k\right)+L_{d}\left(y\left(k\right)-\hat{y}\left(k\right)\right),$$
where $\hat{x}$*(k)* is the *kth* estimated state vector, $\hat{y}$*(k)* is the kth estimated output vector. *A_d_* is the discretized state matrix, $B_{d}$ is the discretized input matrix, and $L_{d}$ is the observer gain matrix.

### 2.4. Deterministic Artificial Intelligence (DAI)

While feedback (PDI) and feedforward control have proven applications and functionality, they both suffer flaws. Feedback (PDI) control contains no analytical solution, as one is equating a derivative and integral to an exact physical value, hence the consistent error and oscillation in steady state of all feedback (PDI) solutions. Adaptive Feedforward control could in theory provide exact solutions, but only with perfect knowledge of all system and environmental parameters, again impossible. deterministic artificial intelligence seeks to provide an exact analytical solution without knowledge of environmental parameters, by enforcing new assumptions on your physical system, a system who’s physics one can characterize exactly. In this case, we focus purely on three degree of freedom rotational control according to Euler’s equations for rotational rigid body motion. Our estimable system parameter of inertia will allow us to enforce desirable physics on our control algorithm via the self-awareness statement displayed in Equation (5) (reference Table 4 for variable definitions in Equations (5)–(7)). whose system matrix is displayed in Equation (6) and regression vector in Equation (7). Equation (6) in particular illustrates the novel method of insuring trajectory tracking, where the control definition embeds the coupled, nonlinear desired trajectory annotated by the subscript, *d*. As estimates converge to actual values, trajectory tracking is assured by actual states converging to desired states.
(5)$$u\equiv \hat{J}{\dot{\omega }}_{d}+\omega_{d}\times \hat{J}\omega_{d}=\left[\Phi_{d}\right]\left\{\hat{\Theta }\right\},$$
where:(6)$$\Phi_{d}=\left[\;\begin{array}{cccccc} {\dot{\omega }}_{x} & {\dot{\omega }}_{y} & {\dot{\omega }}_{x} & -\omega_{y}\omega_{x} & 0 & \omega_{z}\omega_{y}\\ \omega_{x}\omega_{z} & {\dot{\omega }}_{x} & 0 & {\dot{\omega }}_{y} & {\dot{\omega }}_{z} & -\omega_{z}\omega_{x}\\ -\omega_{x}\omega_{y} & 0 & {\dot{\omega }}_{x} & \omega_{y}\omega_{x} & {\;\dot{\omega }}_{y} & {\;\dot{\omega }}_{z} \end{array}\right],$$

And
(7)$$\Theta ={\left\{\begin{array}{cccccc} J_{xx} & J_{xy} & J_{xz} & J_{yy} & J_{yz} & J_{zz} \end{array}\right\}}^{T}\to \hat{\Theta }={\left\{\begin{array}{cccccc} {\hat{J}}_{xx} & {\hat{J}}_{xy} & {\hat{J}}_{xz} & {\hat{J}}_{yy} & {\hat{J}}_{yz} & {\hat{J}}_{zz} \end{array}\right\}}^{T},$$

Incorporating this knowledge with our observer output, we can estimate a $\hat{\theta }$ (the difference in our desired state inertia vs. our current state inertia) via the Moore-Penrose pseudo inverse (the 2-norm optimal solution to our self-awareness statement). This gives an output control that enforces our system inertia to match its actual behavior, thus producing a more robust response regardless of any present disturbances.

Using our learned $\hat{\theta }$, we can enforce an additional control input proportional to $\hat{\theta }$ in addition that of our PID algorithm. While the DAI control can exist on its own, in this paper we only cover the pure PID and hybrid PID/DAI cases.

Existing machine learning techniques in vehicle control attempt to match assumptions about highly non-linear real-world equations of motion and matching control outputs based on environmental experience. Inherently, these methods will be non-robust as the introduction of novel inputs to their schemas will produce undesirable results until the algorithm readapts. For example, if a reinforcement learning model is trained to control an aerial vehicle in laminar flow conditions, the introduction of turbulent conditions will introduce unknown equations of motion that will threaten stability. DAI, in contrast, assumes any estimated control input to produce a desired change in motion is inherently false (as we do not have perfect knowledge of the multitude of effects that can change vehicle dynamics). To counter this problem, it simplifies the error to a change in vehicle parameters (parameters that we know will affect the equations of motion we are seeking to modify, in this case inertia). The “learning” is instantaneous and not necessarily reflective of the true state of the vehicle, but it is reflective of the effective state in the face of unknown perturbations.

## 3. Results

This section will compare combined feedforward and feedback control (FFD + FB) to combined feedforward and feedback control with deterministic artificial intelligence (FFD + FB + DAI). First, we will compare performances of the schemes for a thirty-degree yaw with no disturbances. Next, we will perform the same maneuver with simulated gravitational gradient and drag torques to evaluate the efficacy of each approach to handle the disturbance inputs without their explicit presence in the respective approach. Lastly, we simulate a large one-hundred-degree yaw with perturbations and a change in the craft’s dynamics via a sudden shift in inertia. Visualization, initialization, and data processing code are provided in Appendix B. The exact values for simulation startup and input parameters are provided in Appendix C. Observer and controller gains are listed in Table 5.

### 3.1. Thirty-Degree Yaw

The plots in Figure 2 show that just over one order of magnitude of precision is gained via deterministic artificial intelligence implementation. Exact values of improvement are listed in Table 6. While traditional control settles to $2.1424\times 10^{-4}$ degrees of error, deterministic artificial intelligence hybrid control settles to $1.5147\times 10^{-5}$ degrees of error in a fraction of the amount of time. Additionally, deterministic artificial intelligence runs in 18.2 s while traditional optimal control takes 21.3 s, showing both a boost in speed and accuracy.

### 3.2. 30 Degree Yaw with Perturbations

While the traditional proportional-derivative-integral (PDI) controller’s performance is reduced in the presence of perturbations, we see in Figure 3 and Table 7 that deterministic artificial intelligence’s performance increases, settling to an error of $6.0185\times 10^{-6}$ degrees.

### 3.3. One-Hundred-Degree Yaw Maneuver with Perturbations and Simulated Damage

Lastly, a large maneuver produces largely similar results for deterministic artificial intelligence with a maximum error of $5.4427\times 10^{-6}$ degrees as depicted in Figure 4 and Table 8. Traditional control settles to $6.2142\times 10^{-5}$ degrees of error.

## 4. Discussion

While it was anticipated that the hybrid deterministic artificial intelligence approach would yield optimal results per the work of Smeresky and Rizzo, one surprising outcome was the reduction of error in the presence of persistent excitation. Any level of observer noise improved the precision of the deterministic artificial intelligence controller. In simulation, this can be difficult to model but a real-world implementation of deterministic artificial intelligence would yield continuous noise inputs from sensors and thus more accurate controls.

Both controllers seemed to handle sudden changes in spacecraft inertia (and thus its dynamics), quite well, with minimal effect on the actual control. As displayed in Table 9, mean trajectory tracking error was improved over 91%, while the standard deviation was improved over 97% whilst improving (reducing) control effort by 13%. It should be noted that the observer proportional-derivative-integral (PDI) gains were set quite aggressively and may not be feasible for real life actuators to implement. Further work should be done to explore deterministic artificial intelligence with control signals restricted by the capabilities of real-world actuators.

One final further route to explore would be to implement the $\hat{\theta }\;$ output of the deterministic artificial intelligence controller to update the controller side inertia estimate. This may allow for faster control in the face of damage, along with being a useful diagnostic tool for remote vehicles.

## Figures and Tables

**Figure 1 sensors-22-08723-f001:**
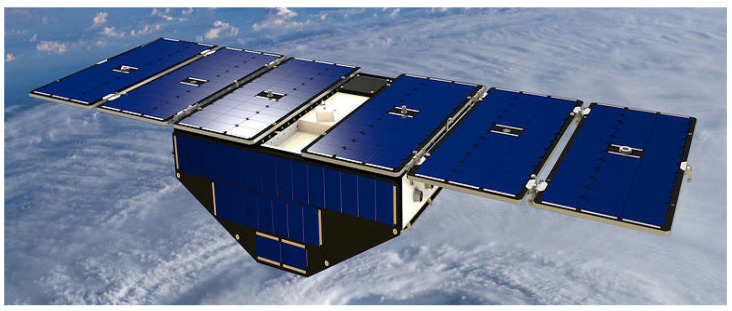
NASA’s Cyclone Global Navigation Satellite System (CYGNSS) mission, a constellation of eight microsatellites, will improve hurricane forecasting by making measurements of ocean surface winds in and near the eye wall of tropical cyclones, typhoons and hurricanes throughout their life cycle. Figure taken from [1] in compliance with NASA’s image use policy [2].

**Figure 2 sensors-22-08723-f002:**
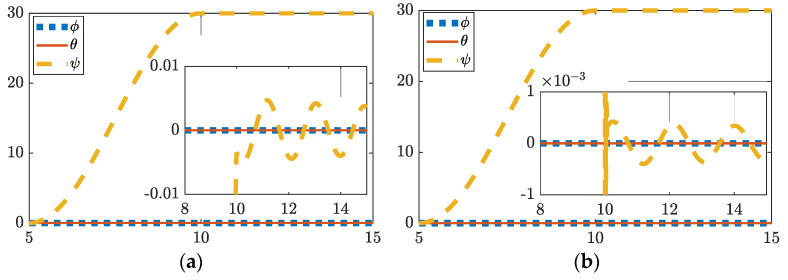
(**a**) Thirty degree yaw feedforward plus feedback (FFD + FB) control, (**b**) thirty degree yaw hybrid deterministic artificial intelligence (DAI) control. Both figures include display of tracking errors in the zoomed-inset graphic. Notice the ordinate scale, respectively of insets in subfigure (**a**) and (**b**) to reveal the relative comparison.

**Figure 3 sensors-22-08723-f003:**
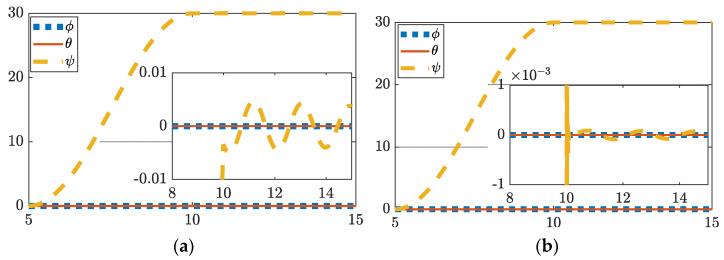
Control with perturbations. (**a**) Thirty degree yaw using feedforward plus feedback (FFD + FB) control; (**b**) thirty degree yaw hybrid deterministic artificial intelligence control. Both figures include display of tracking errors in the zoomed-inset graphic. Notice the ordinate scale, respectively of insets in subfigure (**a**,**b**) to reveal the relative comparison.

**Figure 4 sensors-22-08723-f004:**
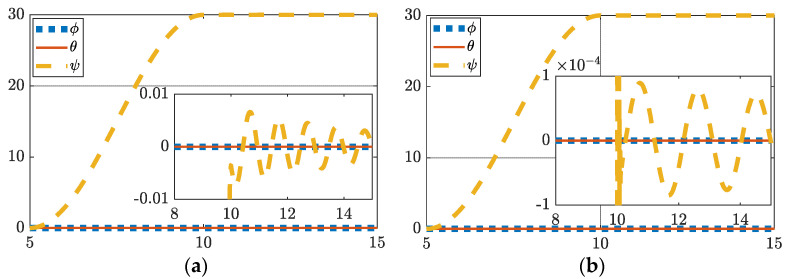
(**a**) One hundred degree yaw feedforward plus feedback (FFD + FB) control; (**b**) one hundred degree yaw with hybrid deterministic artificial intelligence (DAI) control. Both figures include display of tracking errors in the zoomed-inset graphic. Notice the ordinate scale, respectively of insets in subfigure (**a**,**b**) to reveal the relative comparison.

**Table 1 sensors-22-08723-t001:** Definitions of proximal variables for Section 2.1.

Variable	Definition	Variable	Definition
T	Resultant applied torque	ω	Angular velocity (radians/second)
$\dot{H}$	Timed rate of change of $H_{S}$	$H_{S}$	Spacecraft angular momentum

**Table 2 sensors-22-08723-t002:** Definitions of proximal variables for Section 2.2.

Variable	Definition	Variable	Definition
$u_{fb}$	Feedback control signal	$\theta_{d}$	Desired attitude angle
$k_{p}$	Proportional gain	$\omega_{d}$	Desired angular rate
$k_{d}$	Derivative gain	θ	Actual attitude angle
$k_{i}$	Integral gain	ω	Actual angular rate
$e_{\theta }$	Angular position error	*J*	Mass moment of inertia
${\dot{e}}_{\theta }$	Angular velocity error	*dt*	Differential element of time

**Table 3 sensors-22-08723-t003:** Definitions of proximal variables for Section 2.3.

Variable	Definition	Variable	Definition
$x\left(k+1\right)$	State at following timestep	$u\left(k\right)$	Control at present timestep
$A_{d}$	Discretized state matrix	$y\left(k\right)$	Output at present timestep
$\hat{x}\left(k\right)$	Present state estimate	$\hat{y}\left(k\right)$	Present timestep output estimate
$B_{d}$	Discretized input matrix	*k*	Present timestep
$L_{d}$	Observer gain matrix		

**Table 4 sensors-22-08723-t004:** Definitions of proximal variables for Section 2.4.

Variable	Definition	Variable	Definition
u	Total control	${\dot{\omega }}_{x}$	Acceleration about body *x*-axis
$\hat{J}$	Estimated mass moment of inertia	${\dot{\omega }}_{y}$	Acceleration about body *y*-axis
${\dot{\omega }}_{d}$	Desired angular acceleration vector	${\dot{\omega }}_{z}$	Acceleration about body *z*-axis
$\omega_{d}$	Desired angular rate vector	$\omega_{x}$	Angular rate about the body *x*-axis
$\Phi_{d}$	Regression matrix of “knowns”	$\omega_{y}$	Angular rate about the body *y*-axis
$\hat{\Theta }$	Regression vector of “unknowns”	$\omega_{z}$	Angular rate about the body *z*-axis
FFD	Feedforward control	DAI	Deterministic artificial intelligence
FB	Feedback control		

**Table 5 sensors-22-08723-t005:** Observer and controller gains ^1^.

	$K_{p}$	$K_{d}$	$K_{i}\;$
PDI control	1000	10	0.1
Luenberger Observer	10,000	500	0.1

^1^ These gains will remain constant for all data sets.

**Table 6 sensors-22-08723-t006:** Figures of merit for nominal thirty degree yaw ^1^.

Method	Mean Tracking Error (μ)	Tracking Error Standard Deviation (σ)	Control Effort
Feedforward + feedback (PDI)	2.1424 × 10^−4^	2.3 × 10^−3^	2.13 × 10^1^
Hybrid deterministic artificial intelligence	1.5147 × 10^−5^	2.0181 × 10^−4^	1.82 × 10^1^

^1^ Illustration of performance improvement.

**Table 7 sensors-22-08723-t007:** Figures of merit for thirty-degree yaw with perturbations ^1^.

Method	Mean Tracking Error (μ)	Tracking Error Standard Deviation (σ)	Control Effort
Feedforward + feedback (PDI)	2.1537 $\times 10^{-4}$	2.3 $\times 10^{-3}$	2.46 $\times 10^{1}$
Hybrid deterministic artificial intelligence	6.0185 $\times 10^{-6}$	4.3078 $\times 10^{-5}$	2.13 $\times 10^{1}$

^1^ Illustration of performance improvement.

**Table 8 sensors-22-08723-t008:** Figures of merit for thirty-degree yaw with perturbations and simulated damage ^1^.

Method	Mean Tracking Error (μ)	Tracking Error Standard Deviation (σ)	**Control Effort**
Feedforward + feedback (PDI)	$6.2142\times 10^{-5}$	$1.40\times 10^{-3}$	$2.29\times 10^{1}$
Hybrid deterministic artificial intelligence	$5.4427\times 10^{-6}$	$4.0650\times 10^{-5}$	$1.99\times 10^{1}$

^1^ Illustration of performance improvement.

**Table 9 sensors-22-08723-t009:** Percent performance improvements in DAI vs Feedforward+Feedback for thirty degree yaw with perturbations and simulated damage ^1^.

Method	Mean Tracking Error	Tracking Error Standard Deviation	Control Effort
Hybrid deterministic artificial intelligence	91.24%	97.10%	13.10%

^1^ Illustration of performance improvement.

## Data Availability

Data is available by contacting the corresponding author.

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
