# Peer review of "Microsatellite Uncertainty Control Using Deterministic Artificial Intelligence"

_sensors, 2022, doi:10.3390/s22228723_

Round 1

Reviewer 1 Report

The focus of the paper is important, but unfortunately its academic contribution is not strong enough and the work is not presented properly. Please consider the comments below to improve the paper further:

1)      Please provide more accurate and informative title for the paper. The title should reflect the key contributions of the paper. For example, the uncertainties and their control with the AI based control seem the major contribution of the paper.

2)      Abstract of the paper should be improved. The first sentence can state the importance of the content, then the gaps in the corresponding literature. Key contributions of the paper should be expressed clearly and then the major findings of the paper should be provided.

3)      Introduction has provided some background researches and highlighted their advantages and disadvantages. However, critical review of the recent and related works are not quite strong. The corresponding gaps should be emphasized strongly and based on these gaps, the claimed contributions of the paper should be justified.

4)      Please note that the comparisons of the various algorithms should be performed under equal conditions. It requires that the parameters of the algorithm must be chosen proportional to their optimal values. However, to determine the optimal values, analytic solutions of the proposed system is required.

5)      Please improve the equations by adding brief insights about them. Parameters of the equations have not been introduced.

6)      Please specify the kind of uncertainties. They can be internal or external, parametric or non-parametric, constant, characteristic or random. Determining their structures and amounts are challenging in the real time applications. I would suggest you to see this recent and related paper which develops 5-dimensional optimal performance-based policies under the various unstructured uncertainties: Pharmacological, Non-Pharmacological Policies and Mutation: An Artificial Intelligence Based Multi-Dimensional Policy Making Algorithm for Controlling the Pandemic Diseases.

7)      What are the possible problems that the proposed algorithm can face in real time applications? What are the physical, mechanical, electrical and environmental constraints which are unavoidable in real time environments? 

8)      AI is intrinsically stochastic. The paper also mentions a number of uncertainties, disturbances and noises. How can be the algorithm still deterministic?

9)      Starting the introduction with a picture is not quite appropriate for an academic paper.

10)  It is stated that “… in response to unknown sensor noise and sudden changes in craft physical parameters”. Please note that sensor noise is internal non-parametric and craft dynamics change is internal parametric uncertainty. If the AI is not constructed in a way to reflect the structure of the uncertainties, then the terminal learning is just an approximate learning. Please see the suggested paper in Comment 6.

11)  Why are the PID controller and observer introduced?

12)  Details of the deterministic AI is insufficient.

13)  Please improve the results and remove the copied codes.

Good luck with the improvements…

Author Response

1) Please provide more accurate and informative title for the paper. The title should reflect the key contributions of the paper. For example, the uncertainties and their control with the AI based control seem the major contribution of the paper.

Thanks for the suggestion. The title is modified in accordance with the recommendation.

2) Abstract of the paper should be improved. The first sentence can state the importance of the content, then the gaps in the corresponding literature. Key contributions of the paper should be expressed clearly and then the major findings of the paper should be provided.

Thanks for the great suggestion. Abstract is now adjusted accepting this marvelous suggestion.

3) Introduction has provided some background researches and highlighted their advantages and disadvantages. However, critical review of the recent and related works are not quite strong. The corresponding gaps should be emphasized strongly and based on these gaps, the claimed contributions of the paper should be justified.

Thanks.  The introduction is now expressed in accordance with this recommendation.

4) Please note that the comparisons of the various algorithms should be performed under equal conditions. It requires that the parameters of the algorithm must be chosen proportional to their optimal values. However, to determine the optimal values, analytic solutions of the proposed system is required.

Great recommendations, thanks.  You’ll find the updates in section 2 Methods and Materials.

5) Please improve the equations by adding brief insights about them. Parameters of the equations have not been introduced.

Great suggestion.  Proximal tables of equations’ contents have been added in several convenient tables throughout the manuscript.

6) Please specify the kind of uncertainties. They can be internal or external, parametric or non-parametric, constant, characteristic or random. Determining their structures and amounts are challenging in the real time applications. I would suggest you to see this recent and related paper which develops 5-dimensional optimal performance-based policies under the various unstructured uncertainties: Pharmacological, Non-Pharmacological Policies and Mutation: An Artificial Intelligence Based Multi-Dimensional Policy Making Algorithm for Controlling the Pandemic Diseases.

Very interesting suggestion, thanks. Uncertainty designation has been augmented to the previously updated segments particularly as it pertains to the self-awareness statements of deterministic AI. Furthermore, the relationship between machine learning approaches has amplified the literature review’s discussion in this regard.

7) What are the possible problems that the proposed algorithm can face in real time applications? What are the physical, mechanical, electrical and environmental constraints which are unavoidable in real time environments?

Great question, thanks.  The nature of the question and its responsive recommendations have been included in the newly augmented Introduction.

8) AI is intrinsically stochastic. The paper also mentions a number of uncertainties, disturbances and noises. How can be the algorithm still deterministic?

Very natural confusing question, thanks!  The Introduction has been amplified to articulate the disparate treatments.

9) Starting the introduction with a picture is not quite appropriate for an academic paper.

The graphic on the first page is intended to foremost and immediately relay to the reader the nature of the article’s application. Inclusion is deemed a “best practice” for magnifying the professional writing qualities of the manuscript.

10) It is stated that “… in response to unknown sensor noise and sudden changes in craft physical parameters”. Please note that sensor noise is internal non-parametric and craft dynamics change is internal parametric uncertainty. If the AI is not constructed in a way to reflect the structure of the uncertainties, then the terminal learning is just an approximate learning. Please see the suggested paper in Comment 6.

Thanks.  Indeed stochastic approaches necessitate such classification, while the strength of the deterministic approach, particularly the self-awareness statement relegates the learning to necessarily small terms that bely the requirement for such formalization.

11) Why are the PID controller and observer introduced?

Thanks.  The classical approaches are introduced as a benchmark for comparison.

12) Details of the deterministic AI is insufficient.

Indeed.  In response to the issues presented in this review have substantially enlarged the articulation of deterministic AI and its efficacy and weaknesses.

13) Please improve the results and remove the copied codes.

Thanks for the suggestions.  The results are now presented in raw form in section 3, while they are duplicatively presented as “percent performance improvement” in the closing Discussion.

Reviewer 2 Report

This paper discussed the advantage of using deterministic artificial intelligence (DAI) in control design for a rigid Body Motion. However, the reviewer has the following concerns:

1. The section 2.4 "deterministic artificial intelligence (DAI)" needs to be described in detail as well as added more the reference to be cited. Moreover, this AI also needs to be compared with different AI techniques, such as Reinforcement Learning in control design of mobile robots:
https://www.sciencedirect.com/science/article/abs/pii/S0019057822001495; https://onlinelibrary.wiley.com/doi/10.1002/asjc.2830; https://link.springer.com/article/10.1007/s12555-020-0809-7.

2. The posibilty of tracking in the proposed structure should be discussed not only in simulation result but also in theoretical analysis.

3. The method of handling the disturbance should be added in this manuscript.

4. The parameters in object simulation results should be given in this work.

Author Response

  1. The section 2.4 "deterministic artificial intelligence (DAI)" needs to be described in detail as well as added more the reference to be cited. Moreover, this AI also needs to be compared with different AI techniques, such as Reinforcement Learning in control design of mobile robots:
    https://www.sciencedirect.com/science/article/abs/pii/S0019057822001495; https://onlinelibrary.wiley.com/doi/10.1002/asjc.2830;

https://link.springer.com/article/10.1007/s12555-020-0809-7.

  • Great suggestion. Section 2.4 has been updated with more details on DAI and how it differs from existing learning models in robotics

  1. The possibility of tracking in the proposed structure should be discussed not only in simulation result but also in theoretical analysis.
  • Thank you for the great suggestion. Verbiage has been added and emphasis placed on the theoretical expressions, particularly in equation (6).

  1. The method of handling the disturbance should be added in this manuscript.
  • Section 3 Results describes the simulated disturbance inputs (gravity and drag torque). Commentary has been added to illustrate hoe the disturbances were handled, while the strict addition of the disturbances is illustrated in the implementation code in the appendix.

  1. The parameters in object simulation results should be given in this work.
  • Thank you. I have added an Appendix which provides all input parameters to the Simulink reference as Appendix C in the results section.

Reviewer 3 Report

This paper is of value in both therem and applications, and it is well organized, thus it can be accepted by this journal after minor revision:

1. the introduction is not clear enough, please improve the motivation of this paper.

2. some relative lastest papers should be cited.

Author Response

  1. the introduction is not clear enough, please improve the motivation of this paper.
  • Thank you. Please see the revised intro, particularly the last several paragraphs as being motivated by the inherent pitfalls of stochastic models and the previous work by Smereski and Rizzo.
  1. some relative latest papers should be cited.
  • The reworked intro provides further clarity on the latest works driving the motivation behind this paper.

Reviewer 4 Report

I looked upon the article and found it quite interesting. English is good and the topic is highly significant. It is well organized and very formalized, with a strong mathematical calculus and schematics for simulations. I propose it for publication. 

Author Response

Thank you for the detailed review and supportive comments. 

Round 2

Reviewer 1 Report

The paper has been revised and improved partially, but it is still not sufficient enough for an academic paper. Abstract still does not deliver the key properties of the proposed approach and the major contribution of the paper. Introduction still does not cover the crirtical review of the recent and related works with the advanatges and disadvanatges. Method section is still weak. Thet setion should be constructed based on the claimed contributions and theoritical backgorund of the claimed contributions should be justified. Comparing such an approach with the tradional control approaches is not approapriate since the PID control is now in the history. The results should be improved by directly addressing the key findings of the paper. I am not sure why there are insignificant values and large values in the same figures.
